# Hybrid Epoxy Nanocomposites: Improvement in Mechanical Properties and Toughening Mechanisms—A Review

**DOI:** 10.3390/polym15061398

**Published:** 2023-03-10

**Authors:** Anita Białkowska, Mohamed Bakar, Wojciech Kucharczyk, Iwona Zarzyka

**Affiliations:** 1Faculty of Chemical Engineering, University of Technology and Humanities, 26-600 Radom, Poland; 2Faculty of Mechanical Engineering, University of Technology and Humanities, 26-600 Radom, Poland; 3Faculty of Chemistry, Rzeszów University of Technology, 35-959 Rzeszów, Poland

**Keywords:** epoxy resin, hybrid nanocomposites, mechanical properties, toughening, mechanisms

## Abstract

This article presents a review on the recent advances in the field of ternary diglycidyl ether of bisphenol A epoxy nanocomposites containing nanoparticles and other modifiers. Particular attention is paid to their mechanical and thermal properties. The properties of epoxy resins were improved by incorporating various single toughening agents, in solid or liquid states. This latter process often resulted in the improvement in some properties at the expense of others. The use of two appropriate modifiers for the preparation of hybrid composites, possibly will show a synergistic effect on the performance properties of the composites. Due to the huge amount of modifiers that were used, the present paper will focus mainly on largely employed nanoclays with modifiers in a liquid and solid state. The former modifier contributes to an increase in the flexibility of the matrix, while the latter modifier is intended to improve other properties of the polymer depending on its structure. Various studies which were carried out on hybrid epoxy nanocomposites confirmed the occurrence of a synergistic effect within the tested performance properties of the epoxy matrix. Nevertheless, there are still ongoing research works using other nanoparticles and other modifiers aiming at enhancing the mechanical and thermal properties of epoxy resins. Despite numerous studies carried out so far to assess the fracture toughness of epoxy hybrid nanocomposites, some problems still remain unresolved. Many research groups are dealing with many aspects of the subject, namely the choice of modifiers and preparation methods, while taking into account the protection of the environment and the use of components from natural resources.

## 1. Introduction

Epoxy resins are known for their excellent chemical resistance, high specific strength, good dimensional stability and adhesion properties [1,2,3,4,5]. Epoxy resins can be used as liquids or powders, which facilitate their mixing with reactive or non-reactive modifiers. They are used as paints, coatings, adhesives and matrices for structural composites [6]. However, once cured, epoxy resins exhibit a high crosslink density, which leads to increased brittleness, low strain at break, poor impact strength and low resistance to crack propagation. The inherent brittleness associated with the low toughness of epoxy resins constitutes serious limitations for their engineering applications. The structure of diglycidyl ether of bisphenol A epoxy resin is shown in Figure 1.

The modification of epoxy resins to improve the above-mentioned properties has been the subject of intense research interest in the last four decades [7,8,9,10,11,12]. They have been successfully toughened using various suitable modifiers such as synthetic and natural fibers [13,14,15,16,17,18,19], thermoplastics [20,21,22,23,24,25,26,27,28,29,30], liquid rubbers [31,32,33,34,35,36,37,38,39,40], plasticizers [41,42,43,44,45], solid microparticles [46,47,48,49,50,51,52,53] and nanoparticles [54,55,56,57,58,59,60,61,62,63,64,65]. The toughening of epoxy resins was attributed to various mechanisms such as crack-pinning, phase separation, plastic deformation, interpenetrating polymer network formation and exfoliation of nanoparticles. The use of a single modifier for epoxy toughening has been known for many decades, but the preparation of hybrid ternary composites containing two different modifiers is a relatively new concept. It can take advantage of the possible existence of a synergistic effect, leading to a greater improvement in the performance properties of the composite. A correct selection of modifiers could lead to specific interactions between the components of the hybrid composite and therefore to a greater improvement in its properties. The simultaneous use of different reinforcing agents has recently been explored by various researchers as a useful and logical approach to improve the performance properties of epoxy resins and avoid negative side effects such as a reduction in stiffness, strength or temperature glass transition of the matrix.

Due to the very large number of potential modifiers to be used in the preparation of hybrid epoxy composites, this review focuses on nanofillers (such as graphene, carbon nanotubes, montmorillonite, silica, …) and other modifiers such as thermoplastics, liquid rubbers, thermosets, diluents. Very interesting reviews have already been published in recent years on hybrid epoxy nanocomposites [65,66,67,68,69,70]. The manuscript would provide a very significant contribution for industrialists and academics since it features the positive toughening of the DGEBA with two modifiers. This would lead to widening the field of application of reinforced materials or ensuring their safe use under severe conditions.

## 2. Hybrid Epoxy Nanocomposites

Nanoparticles can be classified into different categories depending on their shape, size, morphology and properties. Based on their shapes, nanofillers are classified into three categories [71,72]: nanoparticles with one nanoscale dimension (nanoplatelets), two nanoscale dimensions (nanofibers) and three nanoscale dimensions (nanoparticulates). Carbon nanotubes and halloysite nanotubes are considered as two-dimensional nanofillers, while clay and graphene nanoplatelets are examples of one-dimensional nanofillers. Two-dimensional nanofillers include nanotubes and nanofibers with a diameter lower than 0.1 μm. However, nanometric silica beads belong to the three-dimensional nanofiller group.

It is well known that the dispersion of nanoparticles plays a key role in improving the mechanical and thermal properties of polymer nanocomposites. This can be achieved by the intensive mixing of nanoparticles in the polymer matrix, aiming at the exfoliation and/or intercalation of the former, resulting in a higher contact surface between the constituents.

### 2.1. Hybrid Epoxy Nanocomposites Based on Liquid Rubbers

Epoxy resins were modified with different liquid modifiers such as liquid natural rubbers [73,74,75,76,77,78], epoxidized natural rubber, hydroxyl terminated polybutadiene, amine-terminated polybutadiene, epoxy-terminated polybutadiene and butadiene acrylonitrile rubbers terminated with carboxyl groups, amine groups or epoxy groups.

One of the most promising methods to toughen epoxy resin was the use of liquid rubbers in combination with nanoparticles. Rubber toughening of epoxy resin has been considered for over five decades to be one of the most effective methods for reducing brittleness and increasing other matrix properties. In general, epoxy resins modified with liquid rubbers exhibit improved flexibility and impact resistance. The inclusion of nanofillers in the epoxy matrix along with rubber is expected to improve its strength, stiffness and glass transition temperature, without, however, reducing the decrease in its stiffness, strength or glass transition temperature.

In fact, most of the conducted investigations using nanoparticles with liquid rubbers showed an improvement in epoxy performance properties. The epoxy/rubber nanocomposites have been used in various fields such as adhesives, coatings, transducers and in electromagnetic systems.

Hydroxyl-terminated polybutadiene (HTPB) is a telechelic, translucent liquid rubber which finds a wide range of applications. It is characterized by a low glass transition temperature of about −75 °C, an average molecular weight and a high level of reactive functionality, making it an excellent potential curing modifier for epoxy resin. Various researchers used both nanofillers and HTPB to improve the properties of diglycidyl ether of bisphenol A (DGEBA) epoxy resin [79,80,81,82,83]. Yi et al. [79] studied the properties of a prepared hybrid epoxy (EP) resin based on montmorillonite (MMT) and HTPB with different number average molecular weights. The advantage of using inorganic MMT clay as a reinforcement for a certain number of polymers is associated with their nanometric size and their ability to form intercalated and exfoliated structures (Figure 2). It was shown that the tensile strength and elongation at break of EP/10 wt% HTPB/1 wt% MMT ternary composites were higher than those of epoxy binary blends. Moreover, HTPB with a higher molecular weight led to a more exfoliated nanostructure. It was also confirmed that 40 to 80 °C were the preferred temperatures to obtain nanostructured HTPB/MMT. The toughening of ternary epoxy nanocomposites was attributed to the slow and rapid growth of the cracks and the energy dissipated during the process of pulling out and debonding of nanoparticles from the matrix. The addition of microscale and nanoscale modifiers can lead to different interactions between the two kinds of particles and the matrix, which determine the final morphologies and properties of the hybrid polymer [80].

Marouf et al. [69] confirmed the “nano” effects of silica (<25 vol%) and rubber nanoparticles (>10 wt%) in toughening epoxy resin compared to silica and rubber microparticles at the same loading. Synergistic toughening effects were noted on the fracture toughness parameter G_C_ in epoxy/nanosilica/rubber microparticles, epoxy/nanosilica/rubber nanoparticle composites and epoxy/CNT/rubber nanoparticle composites. The simultaneous incorporation of microscale elastomers/thermoplastics and nanoscale fillers for the modification of epoxy systems was also investigated and confirmed the complicated interaction between the modifier at the micro- and nanoscales and the inherent problems of synergy [80].

Due to its good thermal oxidation resistance, oil resistance and damping properties, acrylic rubber (ACM) particles were combined with montmorillonite (MMT Na^+^) to improve the mechanical and thermal properties of epoxy resin [81]. The tensile strength and modulus were shown to increase and the ductility to decrease with the increasing organic clay content, while rubber had the opposite effects on the epoxy resin properties. However, the ductility of the epoxy matrix was improved without compromising the modulus and strength when both modifiers were used. Relative elongation at break increased maximally in ternary epoxy nanocomposites. However, a synergistic effect was observed on the toughness and tensile modulus of ternary epoxy nanocomposites prepared by the solvent blending of acrylic rubber (ACM) and organomodified montmorillonite (Cloisite30B) [82]. The adhesive strength of an epoxy/ACM blend containing 3 wt% MMT also increased, due to a positive blending effect, resulting from the positive effect of crosslinking and the tearing energy of the adhesive film. The addition of liquid rubbers leads to a softening of the brittle epoxy matrix and decrease in stiffness, which results from the plasticization of the three-dimensional crosslinked network and the increase in the free volume, unlike solid montmorillonite nanoparticles which increase the rigidity of the epoxy matrix [69,82]. Fröhlich et al. [83] reported an improved toughness in hybrid epoxy nanocomposites, but decreased tensile strength, stiffness and glass transition temperature in epoxy composites containing both compatibilized liquid rubber and organophilic fluorohectorite, induced by rubber. Similar results were obtained by Ahmed et al. [84] in an epoxy matrix modified with both reactive rubber nanoparticles (RRNPs) and organically modified montmorillonite (Cloisite-30B). The addition of RRNPs led to a softening of the composites, with a lower stiffness and improved toughness compared to those of the neat resin. In contrast, the use of solid nanoparticles (MMTs) increased the stiffness and lowered the toughness of the nanocomposites. However, the combination of the two optimal percentages of RRNPs and nanoclay in the epoxy matrix improved the flexural stress–strain curve and storage modulus over the neat resin. Shayegan and Bagheri [85] reported, for the first time, a synergistic effect on the fracture toughness parameter (K_C_) of the hybrid epoxy composite containing nanosilica and liquid rubber. The toughness of the ternary composite was higher than those of the binary epoxy systems, due to the obvious positive interactions between the epoxy hybrid constituents.

Graphene has attracted great interest over the past two decades due to its outstanding physical properties. With an excellent balance between their physical properties and low cost, graphene nanoplatelets (GnPs) have become one of the most attractive nanofillers for polymer matrices. They exhibit a very high mechanical strength with 1.0 TPa in Young’s modulus, 130 GPa in ultimate strength, high thermal conductivity of about 5000 Wm^−1^ K^−1^, an extremely high specific surface and non-toxicity. An epoxy resin modified with graphene nanoplatelets exhibited a significant improvement in the tensile modulus and tensile strength [86]. The tensile modulus of the nanocomposite containing 6 wt% of graphene increased by ~23% and its ultimate tensile strength decreased by ~54% compared to the pure matrix. The increase in epoxy stiffness was explained by a high tensile strength and high specific area of GnPs [87].

Still, with the aim of achieving an improvement in the mechanical properties of the matrix, another rubber, namely acrylonitrile butadiene terminated by carboxyl (CTBN), was used with graphene nanoplatelets (GnPs) with different diameters [88]. The addition of 3 wt% GnP with a diameter of 5 μm in 10 wt% CTBN has been shown to yield a ternary GnP/CTBN/epoxy nanocomposite with enhanced toughness and thermal conductivity, combined with a comparable stiffness to that of pure resin. The same mechanism as that of the work of [85] can be used to explain the increase in toughness of the epoxy resin.

A very successful approach that has been widely used by various researchers to toughen diglycidyl ether of Bisphenol A (DGEBA) modified with nanoparticles involved the addition of a liquid copolymer of acrylonitrile butadiene terminated by carboxyl (CTBN), amine (ATBN) and epoxy (ETBN) groups [89,90,91,92,93,94,95,96,97,98]. It is important to emphasize that the dispersion of nanoparticles in epoxy/rubber plays a crucial role in improving the properties of the material. However, it was found that well-dispersed organoclay (Cloisite 30B) in a CTBN/epoxy system with a suitable preparation method leads to an effective improvement in the tensile strength and modulus of an epoxy ternary composite [89]. The tensile strength of the hybrid nanocomposite was effectively improved when the MMT was first dispersed in the CTBN rubber phase. In addition, the degree of Cloisite–Na intercalation in the blend and with it, the performance properties of the composite were further increased by the compatibilizing effect of the curing agent.

Using high pressure and direct mixing methods, Liu et al. [90] showed that the fracture toughness parameters (K_C_ and G_C_) maximally increased by 2.2 and 7.6 times, respectively, at 6 phr organoclay (Nanomer I.30E) and 20 phr CTBN loading compared to the pristine resin. A positive superposition effect was obtained on the fracture toughness of the epoxy matrix modified with both rubber and organoclay. The ultimate strength, yield strength and glass transition temperature of the epoxy resin modified with rubber and organoclay have also been improved. Similar findings were obtained by Chonkaew et al. [91] regarding the improvement in the mechanical and tribological properties of epoxy resin using CTBN and organomodified MMT (Nanomer 1.31 PS). It should be noted that a small amount of 1 phr of nanoclay was sufficient to optimally improve both the mechanical and tribological properties of the epoxy matrix modified with 2.5 phr CTBN. It was also reported that epoxy/nanoclay/CTBN ternary nanocomposites prepared via sonication method exhibited the best tensile strength, due to a uniform and stable morphology as well as a good dispersion of nanoclay (Nanomer 1.28E) in the epoxy/CTBN system [92]. Cavitation of CTBN, connected with the formation of voids, debonding of nanoparticles and plastic deformation of the matrix are processes that require additional energy which will count in increased toughness [88,93].

Moreover, it was shown in a separate study that the fracture toughness parameters (K_C_ and G_C_) of ternary epoxy/CTBN/SN composites were higher than those of epoxy/SN and epoxy/CTBN binary systems [94]. The K_C_ value of the hybrid composite containing 15 wt% CTBN and 5 wt% SN more than doubled compared to the virgin matrix with a distinct positive synergistic effect between the CTBN and SN nanofillers on the fracture energy, G_C_. Liang and Pearson [95] confirmed the positive hybrid effect in their epoxy composites containing 3 wt% SN (20 nm in diameter) and 18 wt% CTBN. However, a negative hybrid effect on the fracture toughness parameter, G_C_, was also obtained with more than 5 wt% SN in epoxy, most probably due to particle agglomeration.

Metal oxides such as Al_2_O_3_, ZnO and TiO_2_ are generally used to solve the problems of corrosion resistance, stiffness, toughness and conductivity of epoxy systems [96]. Alumina nanoparticles (Al_2_O_3_) were also used in combination with CTBN liquid rubber to increase the fracture toughness of an epoxy resin. Hosseini and his research group [97] prepared hybrid epoxy nanocomposites with an optimum content of CTBN (15 phr) and different amounts of Al_2_O_3_ nanoparticles (0–10 phr). Hybrid nanocomposites containing 15 phr of CTBN and 7 phr of Al_2_O_3_ exhibited the maximum values of the critical stress intensity factor (K _C_) and critical strain energy release rate (G_C_).

In a comprehensive review, Kausar [98] compared the effects of various nanofillers such as graphene, carbon nanotube, carbon black, nanoclay, silica and other nanoparticles on the properties and structures of rubber-toughened epoxy resins. These nanofillers have enabled the development of rubber-reinforced epoxy nanocomposites with improved performance and their use in adhesives, coatings, anticorrosives and radiation shielding materials for military, aircraft and aerospace applications. The nanocomposite materials offer the advantages of light weight, high strength, high toughness and corrosion resistance. Zewde et al. [99] studied the effect of functionalized carbon nanotubes (fCNTs) and micron-sized rubber particles on the properties of epoxy resin. The nanotubes were mixed with carboxyl-terminated butadiene acrylonitrile-toughened epoxy (CTBNTE) by shear mixing and sonication. The fracture toughness of the epoxy/CTBN/fCNT ternary composites was higher by approximately 200% compared to the neat epoxy matrix, 20% and 110% compared to the epoxy/fCNT and epoxy/CTBN blends, respectively. The confirmed synergistic effect of fCNT nanofillers and rubber particles was attributed to plastic deformation, stress whitening and the overall rougher surface of the ternary composites compared to the pristine resin. Similar results were obtained by Hsieh et al. [100] with a significant improvement in the fracture toughness (G_C_) of an epoxy matrix due to the synergistic effect of silica nanoparticles (SNs) and rubber particles. The critical strain energy release rate parameter, G_C_, of the epoxy resin increased from 77 to 191 J/m^2^ and 671 J/m^2^ by adding 15 wt% SN and 9 wt% of rubber particles, respectively. However, it increased up to 965 J/m^2^ for the ternary epoxy nanocomposites containing both SNs and rubber microparticles. The improvement in the toughness of the epoxy matrix containing soft and solid particles is explained by the same mechanism reported previously [85,88,93].

Similarly to CTBN, liquid butadiene acrylonitrile terminated with amine groups (ATBN) was also used to toughen epoxy nanocomposites [101,102,103]. It was shown that the addition of 1.5 wt% nanoclay to an epoxy composition modified with 5 wt% ATBN led to synergistic effects on the mechanical behavior [101]. A similar improvement in the epoxy resin mechanical properties was noted in the case of epoxy resin modified with a combination of ATBN and organomodified nanoclay (Nanobent ZW1) [102]. Ternary epoxy composites containing 1 wt% or 2 wt% Nanobent and ATBN showed improved mechanical properties in relation to the unmodified EP. The impact strength (IS) and the critical stress intensity factor (K_C_) values of EP containing 1 wt% nanoclay increased approximately by 200% and 75%, respectively, in relation to the neat EP. In another study, the matrix was modified with Nanobent ZW1 and ATBN with different amine equivalent weights (ATBN-16 with 18% acrylonitrile and ATBN-21 containing 10 wt% acrylonitrile) [103]. A ternary composite containing 2 wt% nanoclay and 5 wt% ATBN exhibited a synergistic effect on the tensile adhesive strength toward the binary systems. The significant enhancement in the properties was explained by the reaction that took place between the amine groups of the modifier or hardener and the unreacted part of the cured epoxy matrix. Figure 1 shows how the incorporation of a rubber can change the morphology of a composite to a more elongated one with plastic deformation, thus explaining the increase in the mechanical properties of the matrix.

Figure 2 shows the effect of the reactive rubber and Nanobent ZR1 content on the impact strength (IS) and critical stress intensity factor (K_C_) of an epoxy composite. It is seen that the hybrid composite can be prepared by fixing the content of one modifier with the highest value and by varying the amount of the second modifier.

Kong et al. [104] prepared a series of epoxy composites based on organomodified montmorillonite (oMMT) and epoxy-terminated acrylonitile butadiene copolymer (ETBN) and tested their mechanical performance. The addition of both oMMT and ETBN led to a significant improvement in the epoxy matrix damping properties while maintaining the high tensile strength. The improvement in the damping performance due to the incorporation of ETBN-intercalating oMMT nanocomposites has been confirmed by Mao et al. [105].

Due to its reactive groups, polysulfide is considered, along with liquid rubbers such as CTBN, ATBN and hydroxyl-terminated polybutadiene, as the most effective toughening agent for epoxy resins. The reaction between the thiol and the epoxy groups is expected to enhance the flexibility and impact strength (IS) of the epoxy matrix but decrease certain properties such as the glass transition temperature (Tg) and modulus. Pristine and carboxyl-functionalized multiwalled carbon nanotubes (p-MWCNTs and f-MWCNTs) in addition to polysulfide were used to prepare an amine-cured epoxy resin with improved mechanical properties [106]. The incorporation of 0.1 wt% f-MWCNTs to the epoxy/polysulfide blend significantly improved the tensile and impact resistance of the epoxy nanocomposite. The Tg of the f-MWCNT/epoxy hybrid composites increased by 9.8 °C compared to the neat epoxy matrix. The improvement in the mechanical properties was attributed to the strong interfacial interactions between the f-MWCNTs and the epoxy matrix. The increase in the properties was explained by the plasticization of the cured epoxy by PSf, and most likely to increased matrix crosslinking [107].

Other liquid rubbers with specific physical properties have also been used in combination with nanoparticles to improve the properties of epoxy nanocomposites [108,109,110,111]. Tang et al. [108], for example, investigated the properties of epoxy resin modified with rigid silica nanoparticles (SNs) and phase-separated submicron soft rubber particles. Ternary epoxy composites showed a good balance between the stiffness, strength and fracture toughness compared to single-phase particle binary systems. The fracture toughness improvement in the ternary epoxy composites was attributed to the formation of a large plastic zone around the crack tip, resulting from the dual addition of rigid and soft particles.

Epoxidized natural rubber (ENR) has been used by various researchers to modify epoxy resins, for its polarity and good compatibility with several polar polymers [109,110,111,112]. Leelachai et al. [109] modified an epoxy resin cured with cycloaliphatic polyamine by using epoxidized natural rubber (ENR) with silica nanoparticles (SNs). The addition of ENR resulted in a significant improvement in the fracture toughness parameter (K_C_) with a decrease in the glass transition temperature (T_g_,) and Young’s modulus. However, the addition of SNs led to a modest improvement in the toughness and T_g_ but a significant increase in the modulus. The ternary hybrid epoxy composites exhibited improvements in the K_C_, stiffness and Tg. The maximum improvement in the fracture toughness, including a synergistic effect, was obtained with an epoxy matrix containing a defined amount of SNs and ENR. Morphology analysis revealed the existence of the cavitation of rubber particles together with matrix shear yielding, particle debonding and an increase in the damage zone size, which was associated with the dissipation of more energy during the crack propagation. Recently, Kam et al. [110] studied the effect of different vulcanized rubbers on the properties of epoxy resin filled with graphene nanoplatelets (GNPs). The epoxy matrix was modified with natural rubber (NR), liquid natural rubber (LNR) and recycled natural rubber (rNR). An epoxy/GNP nanocomposite filled with 5 vol% NR showed the highest values of flexural strength, flexural modulus and fracture toughness (K_C_), as compared with LNR and rNR, due to better interfacial adhesion between the epoxy/NR/GNP system. Similar results were obtained on the properties of epoxy resin modified with natural rubber and graphene nanoplatelets [111].

The effect of preformed powdered rubber (PR) nanoparticles and chemically reduced graphene oxide (CRGO) was also evaluated on the mechanical and thermal properties of epoxy resin [112]. The inclusion of CRGO improved the fracture toughness, stiffness and thermal properties of epoxy resin, while the addition of PR resulted in a significant improvement in the fracture toughness but large reduction in the thermal stability and stiffness of the polymer matrix. The results confirmed the existence of a good balance between the stiffness, strength, fracture toughness, thermal stability and glass transition temperature in ternary epoxy hybrid composites. The shear bands formed in the matrix most likely induced by the stress transfer, combined with the nanoparticle pull-out, can partly explain the improved properties of the hybrid epoxy nanocomposites [94,100,110].

A very interesting approach to toughen epoxy resin was the use of core–shell rubber (CSR) particles, which constitute the specific group of rubber of the effective toughening agent for epoxy resins. The particles are generally based on an elastomeric core such as a homopolymer of butadiene or a styrene–butadiene copolymer and a thin epoxy-compatible shell. The particular advantage of CSR particles compared to liquid rubbers is their compatibility with EPS. Tsang and Taylor [113] solved the brittleness problem of adhesives based on the anhydride cured epoxy thermosetting polymer by using CSR particles and silica nanoparticles (SNs). The fracture energy parameter, G_C_, of the modified epoxy was measured at ambient and low temperatures (−40 °C and −80 °C). The results showed that hybrid composites with 10 wt% SN and 10 wt% CSR particles exhibited the maximum improvement in the G_C_, but with no dual strengthening of the matrix, most probably due to the large size of the modifier particles. A similar study was performed with epoxy resin modified with CSR combined with SNs but different results improved the properties and mechanisms [114]. The soft shell was a copolymer synthesized from caprolactone and meso-lactide which was compatible with the epoxy system. The incorporation of 2 wt% of nanofillers resulted in a 40% increase in the impact resistance of the epoxy, while the glass transition temperature decreased slightly. Crazing, formation of microcracks and debonding of nanoparticles from the matrix were the main causes leading to the improved fracture toughness. Soft core–shell rubber (CSR) nanoparticles have also been used with multiwalled carbon nanotubes (MWCNTs) and SiO2 nanoparticles as rigid particles to toughen a bisphenol A-based epoxy cured with an amine hardener [115].

The MWCNT modifiers have attracted great interest as they generally lead to excellent mechanical, thermal and electrical properties in polymer composites. Furthermore, CSR particles have the ability to promote compatibility between incorporated modifiers with the matrix. Epoxy hybridization with CSR and SiO_2_ resulted in a simultaneous improvement in the tensile and fracture properties. The fracture toughness parameter, K_C_, and the fracture energy parameter, G_C_, of hybrid systems containing 5 wt% CSR and 10 wt% SiO_2_ increased to a maximum of about 220% and 900%, respectively, compared to the virgin polymer matrix. However, an increase of 165% was reported for the K_C_ of the epoxy hybrid based on 0.075 wt% MWCNTs and 10 wt% SiO_2_. The so-called core–shell impact modifier particles (CSIMPs) were added with multiwalled carbon nanotubes (MWCNTs) to improve the toughness and tensile properties of the epoxy resin [116]. The soft particles consisted of a poly(butyl acrylate-allyl methacrylate) elastomeric core and a poly(methyl methacrylate-glycidyl methacrylate) shell. It was demonstrated that the highest fracture toughness of the ternary composite was achieved with the addition of 3 wt% MWCNTs and 3 wt% CSIMPs, while the maximum values of the tensile strength and modulus were obtained with same amount of MWCNTs and a lower content (only 1.03 wt%) of the CSIMPs. The improvement in the fracture toughness of the ternary epoxy composites was attributed to deflection/crack arrest as well as expanded plastic deformation around the crack tip induced by the combination of rigid and soft particles. In addition to the processes related to solid particle bridging, the crack deflection and increase in the crack path caused the observed reinforcement [117]. However, Zhu et al. [118] used core–shell rubber (CSR) with zinc oxide-functionalized multiwall carbon nanotubes (ZnO-MWCNTs) to improve the properties of epoxy resin. The functionalized multiwalled carbon nanotube is expected to further increase the properties of the matrix compared to the virgin modifier [106]. MWCNT modifiers have attracted great interest as they generally lead to excellent mechanical, thermal and electrical properties in composites. Additionally, CSR particles have the ability to promote compatibility between incorporated modifiers with the matrix. The obtained results confirmed that the fracture toughness parameter, K_C_, of the hybrid epoxy nanocomposite was higher than those of the matrix binary systems, due to the synergistic effect associated with the shear band formation along with the crack deflection. Similar impressive toughening was achieved by Mehrabi-Kooshki and Jalali-Arani [119] with hybrid epoxy nanocomposites based on CSR particles and graphene oxide (GO). As expected, the tensile strength and modulus of the epoxy systems modified with CSR decreased with the increasing CSR content, due to matrix flexibilization. With the incorporation of only 0.05 phr GO to the neat epoxy and epoxy/rubber blend, the hybrid epoxy composites exhibited a significant improvement in the tensile strain at break compared to the virgin matrix and binary systems. The strain at the break increase resulted mainly from the rubber particle cavitation and the reduction in the epoxy crosslinking density. However, when GO was replaced by silica nanoparticles (SNs), hybrid epoxy nanocomposites containing 4 vol% of SNs and 7 vol% CSR showed a fracture energy five times greater than that of the unmodified epoxy resin with no reduction in the tensile modulus [120]. The increase in the fracture energy of the matrix was explained by the shear yielding observed at the crack tips in the epoxy matrix and CSR cavitation and void growth. Interestingly, the observed lack of SN debonding acted as shear band stoppers. However, due to different constituents and their relevant possible interactions, it can be emphasized that there is no exclusive interpretation of the toughening mechanisms in epoxy composites based on CSR and SNs.

Most of the hybrid epoxy resin composites based on solid nanoparticles and liquid modifiers included nanoparticle debonding and void formation yielding of the polymer matrix. The created voids by the microparticles slow down the crack propagation and facilitate the plastic deformation of the epoxy matrix. Table 1 summarizes epoxy nanocomposites modified with rubbers and shows the synergistic effects on the matrix properties.

### 2.2. Hybrid Epoxy Nanocomposites Based on Polyurethanes

Due to their specific mechanical, physical, biological and chemical properties, polyurethanes (PUs) are a class of versatile materials which can be used in different applications. Various studies concerned the toughening of epoxy resins by using polyurethanes and nanoparticles [121,122,123,124,125,126,127,128,129]. Jia et al. [121,122] demonstrated that the mechanical and thermal properties of interpenetrating epoxy/polyurethane polymer networks (IPNs) modified with organophilic montmorillonite (oMMT) were superior to those of pure PUs and PU/EP IPNs, due to an increased degree of exfoliation of the oMMT and better compatibility between PUs and EP. A strong interaction between the oMMT and the EP/PU IPNs was confirmed with an increase in hydrogen bonding at the interface between the oMMT and the EP/PU blend. In a separate work, it was shown that the effective dispersion of oMMT in an EP/PU IPN system resulted in a synergistic effect in the tribological performance of EP/PU nanocomposites [123]. The obtained results also indicated that the thermal degradation temperature of the IPNs was attenuated by the addition of organically modified clay to the epoxy matrix. Polyurethanes with various structures and mechanical properties were also investigated by other researchers [124,125]. The maximum improvement in impact resistance was obtained with a composite containing 2 wt% nanoclay and 10 wt% PUs synthesized from polyethylene glycol with a molecular weight of 400 g/mol (PU400) and with 1 wt% nanoclay and 15 wt% PU400, corresponding, respectively, to 110% and 75% with respect to the unmodified epoxy resin [124]. However, a 10-fold increase in the flexural strain at break was observed for the composite containing 15 wt% polyoxypropylene diol-based PUs with a molecular weight of 1000 g/mole and 2 wt% nanoclay, including a synergistic effect. It is well known that the flexible polyurethane chains form interpenetrating polymer networks with the epoxy matrix with increased elasticity, while the solid nanoparticles increase the strength and rigidity of the matrix. Moreover, it is expected that chemical reactions occur between the hydroxyl groups of the EP and the isocyanate groups of the PUs (Figure 3). Hybrid composites with improved mechanical properties will be obtained due to the increased yielding of EP and nanoclay exfoliation [125].

The impact strength (IS) and critical stress intensity factor (K_C_) of the epoxy matrix containing 1 wt% and 2 wt% of nanoclay (Nanobent ZR1) are shown in Figure 3 as a function of commercial linear polyurethane (Desmocap 12). Although both parameters express the resistance to crack propagation, they show different trends as a function of the polyurethane content. The explanation would be related to the order of addition of the modifiers in the matrix.

Hydrogen bonds can also form at the interface between the organomodified montmorillonite (MMT) and the epoxy/polyurethane interpenetrating polymer network (EP/PU IPN), leading to a further improvement in the properties of the polymer blend and that of the epoxy matrix (Figure 4).

Moreover, it should be remembered that modified montmorillonite (Cloisite 30B) contains quaternary ammonium ions with two hydroxyethyl groups which are able to react with the isocyanate groups of polyurethane, as shown in Figure 5. This reaction can be confirmed by the FTIR measurement with a decrease in the peak height of the OH groups at 3300 cm^−1^.

Recently, much attention has been paid to the toughening of epoxy resin using nanoparticles in combination with vegetable oil-based polyurethanes [126,127,128,129]. Due to their remarkable importance as an environmentally friendly component, vegetable oils have attracted considerable attention as a potential alternative to polyols for the synthesis of polyurethanes. Although some studies on polyurethane nanocomposites are available, works on vegetable oil-based polyurethane nanocomposites are currently limited. Polyurethanes based on vegetable oil are known for their very good corrosion resistance, good electrical insulation as well as their shape memory. They are also inexpensive, available in large quantities and biodegradable. Li [126] was one of the first to modify interpenetrating polymer networks (IPNs) prepared from epoxy (EP) resin and castor oil-polyurethane (PU) by using different amounts of organophilic montmorillonite (oMMT). Instrumental analysis of the nanocomposites confirmed a uniform dispersion of the nanoparticles in the IPNs with the formation of intercalated or exfoliated microstructures. Differential scanning calorimetry results showed that oMMT promoted the compatibility between the EP and PU phases accompanied by an improvement in the glass transition temperature of the oMMT/EP/PU ternary nanocomposites with the increasing oMMT content. The mechanical properties and thermal analysis indicated that oMMT and the IPN systems exhibited a synergistic effect on improving the mechanical and thermal properties of the pure EP. It appeared that the formation of cavitations and small cracks contributed to the improvement in the EP properties. However, it has been shown that the use of multiwalled carbon nanotubes (MWCNTs) to modify the properties of IPN castor oil-based PU/epoxy resin led to a significant improvement in the matrix properties [127]. The tensile strength of epoxy composites containing 0.1 wt% and 0.7 wt% MWCNTs increased by more than 30% relative to the matrix. The impact strength of the composite with 0.3 wt% MWCNT content increased by approximately 55% compared to the virgin epoxy resin, while the thermal decomposition temperature decreased slightly, due most probably to the strong interaction between the nanoparticles and epoxy/PU matrix. Another vegetable oil (soybean oil) was used by Xu et al. [128] to prepare a polyurethane (PU) which was then mixed with an EP to form IPN structures. The ternary epoxy/PU composites containing oMMT exhibited an increased thermal stability as well as a synergically enhanced tensile strength and tensile modulus. The tensile strength and modulus of the PU/EP IPNs increased maximally by approximately 65% and 390% with 4 wt% and 6 wt% of oMMT, respectively. In a separate study, the tensile strength and scratch hardness of the epoxy resin were significantly improved, and the impact resistance increased slightly when the polyurethane based on *Mesua ferrea* L. seed oil (MFLSO) was used with organically modified bentonite to improve the matrix properties [129]. The thermal stability of the nanocomposites increased by about 40 °C.

The last decades have seen a very significant growth in interest in the use of bioderived products, which has been driven by the need to replace petroleum-based materials and produce eco-friendly materials. Recent research has focused on the use of environmentally friendly and non-toxic routes for the synthesis of polyurethanes, which avoid the use of diisocyanates. The procedure offers a number of advantages, including the use of environmentally friendly, inexpensive and biobased components. However, there remains a great challenge to produce epoxy hybrid nanocomposites based on renewable resources with improved performance properties and taking into account environmental protection. Despite the various studies conducted so far to assess the fracture resistance of epoxy nanocomposites, some issues still remain unclear.

Białkowska et al. [130] prepared and evaluated the mechanical properties of an epoxy resin by using condensation segmented nonisocyanate polyurethane (NIPU) and nanoclays (Nanobent) and analyzed the correlation between morphology and blend performance. The highest values of the impact strength, flexural strength and critical stress intensity factor were obtained with epoxy hybrid composites containing 10 wt% NIPU and 1 wt% nanoclay. As in the case of other studies on conventional isocyanate-based PUs [25,26], it was confirmed that the improvement in the mechanical properties of EP was due to the formation of an epoxy/NIPU network, an extensive matrix yielding with the formation of stretched morphology, but with no grafting reaction.

A green polyurethane (NIPU) was synthesized from sunflower oil via an environment-friendly and non-toxic route and used as a modifier for epoxy resin containing amine-functionalized graphene oxide (af-GO) [131]. The properties of the nanocomposites have been shown to increase with the increasing af-GO content, due to the strong hydrogen bonding and interactions between af-GO with the polymer network. The prepared hybrid epoxy nanocomposites exhibited a good balance between mechanical, chemical and thermal properties. These results highlight the potential of this environmentally friendly approach to prepare renewable NIPUs and high-performance nanocomposite materials.

The occurrence of chemical reactions between components in hybrid epoxy nanocomposites is possible due to the presence of urethane linkage in the polyurethane structure. Therefore, the order of the addition of components is expected to have a crucial effect on the properties of epoxy composites [132,133]. Cheng et al. [132] studied the component order of addition on selected properties of epoxy (EP) resin cured with triethylenetetramine (TETA) and modified with polyurethane (PU) synthesized by reacting polytetramethylene ether glycol (PTMG), isophorone diisocyanate (IPDI) in the presence of dibutyltin dilaurate (DBTDL). The results confirmed the role of the components in the mixing order, in that the best physical, mechanical and thermal properties were obtained when the epoxy matrix was first mixed with PTMG, IPDI, DBTDL, before adding TETA. Similar findings were obtained with two different mixing sequences and their effects on the properties of epoxy ternary nanocomposites [133]. The mechanical and thermal properties of epoxy/organoclay/polyether polyol ternary nanocomposites depended largely on the mixing sequence of the constituents. The results showed that the impact strength values of the ternary nanocomposites were higher when the polyether polyol was mixed with the curing agent followed by its incorporation into the EP/MMT mixture. This was explained by the possible interactions between all the components of the prepared ternary nanocomposites. The importance of the order of the incorporation of the constituents was confirmed in our work on the properties of an epoxy resin modified with montmorillonite (MMT) and conventional polyurethane (PU) [134]. The best mechanical properties were exhibited by nanocomposites in which prepared PU was first mixed with EP before the incorporation of MMT (Cloisite 30B). Structural analysis indicated the formation of a non-grafted interpenetrating polymer network structure. The improvement in the properties was explained by the good interactions between the MMT and the polyurethane, possibly via hydrogen bonding, which, depending on the degree of dispersion of the stratified montmorillonite in the matrix, tends to improve its intercalation [122,124]. The order of the incorporation and mixing of the epoxy nanocomposite constituents are shown in Figure 6.

The addition of flexible polyurethane chains to an epoxy matrix has been shown to result in the formation of an interpenetrating polymer network (IPN) structure with improved ductility but also toughness with hydrogen bonding and grafting reactions. Moreover, the incorporation of solid nanoparticles to the IPN system contributes to the further reinforcement of the polymer matrix.

Table 2 presents epoxy nanocomposites modified with polyurethanes with a synergistic effect on the mechanical properties.

### 2.3. Hybrid Epoxy Nanocomposites Based on Thermoplastics

The concept of incorporating both thermoplastic polymers and nanoclay in the epoxy matrix has been advanced as an efficient method to produce composites with improved mechanical and thermal properties. Thermoplastics have been intensively studied for about four decades as toughening agents because they generally do not cause a significant reduction in the modulus, yield strength and glass transition temperature, as in the case of liquid rubbers. Various ductile thermoplastics were investigated as an alternative to reactive rubber for improving the mechanical properties of the epoxy resins [135,136,137,138,139,140,141,142,143,144,145,146,147,148,149,150,151,152,153,154,155,156]. When mixed with an engineering polymer, the epoxy matrix can form an interpenetrating network which would prevent the agglomeration of the nanoparticles and thus allow their unform dispersion in the blend.

Although they have a lower strength and toughness than engineering plastics, commodity plastics have been employed in combination with nanoparticles to reinforce the brittle thermosetting epoxy material. Park and Jana [135] were the first to study the modification of a diglycidyl ether of bisphenol A (DGEBA) epoxy resin with polymethyl methacrylate (PMMA) and nanoclay. Mixtures of aromatic and aliphatic DGEBA resins were prepared with organically modified montmorillonite (Cloisite 30B) and PMMA. The results confirmed that clay nanoparticles were fully exfoliated in the three-phase composites when the ratio of epoxy to clay was 10, resulting in a significant improvement in the tensile and impact strengths. However, Hernandez et al. [136] confirmed that the best dispersion of nanoparticles in ternary epoxy composites does not lead necessarily to the highest values of the mechanical properties. A synergistic effect was confirmed in a hybrid composite based on the same matrix and PMMA, but with a different nanoclay (Nanobent ZS1) using mechanical mixing followed by additional ultrasonic mixing [137]. The flexural strain at break and flexural energy to break of the composite containing 1 wt.% MMT and 5 wt% PMMA were higher compared to the epoxy binary systems. The same finding was obtained for the impact strength and brittle fracture energy of composites based on 2 wt% MMT and 5 wt% PMMA. The results demonstrated the formation of hydrogen bonding between PMMA and the epoxy matrix, according to Figure 7. The improvement in the properties was explained by the specific interaction between the polymer modifier and the matrix but also with the uniformly dispersed and well-embedded nanoparticles in the polymer blend [138]. Rudresh et al. [139] used the same blending procedure and obtained an improved tensile strength and Young’s modulus of hybrid epoxy nanocomposites containing 6 wt% PMMA and 4 wt% of modified halloysite nanoclay tubes (HNTs). The fracture toughness parameter, K_C_, of the hybrid nanocomposite increased by approximately 95%, most probably due to the positive effects generated from the combination of two dissimilar toughening mechanisms, but also the prepared suspension of nanoparticles which is supposed to ensure the uniform dispersion and exfoliation of the latter.

Poly vinyl chloride (PVC), which belongs to the important group of commodity thermoplastics with interesting properties, was also used to toughen epoxy nanocomposites. Kaushal et al. [140] studied the properties of the nanocomposites of epoxy/PVC with different amounts of organomodified montmorillonite (oMMT) and found that the best mechanical properties were attributed to ternary epoxy composites with 5 phr PVC and 5 phr oMMT.

Another alternative to toughen epoxy resins was the use of tough engineering thermoplastics, such as polysulfone (PSU), polyethersulphone (PES), poly (ether ether ketone) (PEEK), polyetherimide (PEI), acrylonitrile–butadiene–styrene copolymer (ABS), polycarbonate (PC), poly(styrene-co-acrylonitrile) (SAN) and high impact polystyrene (HIPS) [141,142,143,144,145,146,147]. Wang et al. [141] used both polysulfone (PSU) and graphene oxide (GO) to toughen DEGBA epoxy resin. The resistance to crack propagation expressed by the fracture toughness parameter, K_C_, and the elongation at break of epoxy ternary composites containing 5 phr PSU and 0.2 phr GO were maximally increased by 90% and 55%, respectively, compared to the pure matrix, while the tensile strength, Young’s modulus and thermal stability were not affected. The current strategy of using both PSU and GO as dual toughening agents is promising for effectively toughening epoxy resins without sacrificing mechanical and thermal properties. A similar improvement in mechanical properties was obtained by Rajasekaran et al. [142] with ternary epoxy composites based on oMMT and PSU. Introducing 5 wt% oMMT in the modified epoxy/PSU IPN system improved both the tensile and flexural strength compared to the virgin matrix. A maximum increase in the impact strength of 56% was exhibited by the hybrid composites containing 8 wt% PSU and 5 wt% oMMT. The improvement in the mechanical properties of the ternary composites was explained, on the one hand, by the formation of a flexible IPN structure within the system, but also by the effective interaction between the nanoclay, the PES and the epoxy matrix. Studies undertaken by Wang et al. [143,144] concerned the preparation of epoxy/polyethersulfone mixtures reinforced with organoclay montmorillonite (Nanocor 1.30E). Ternary composites prepared with the environmentally friendly melting method showed high tensile properties, a high fracture toughness, K_C_, and good thermal performance compared to the solvent method. The dual addition of PES and organoclay resulted in a synergistic effect on the fracture toughness of the epoxy resin, regardless of the method used [143]. The K_C_ of the EP/PES blend modified with 1 wt.% organoclay increased by about 98% compared to the neat epoxy matrix, while it was higher by 49% and 58% compared to the EP/PES and EP/nanoclay systems, respectively. A different toughening mechanism has been proposed compared to the above studies, and which would involve the debonding of the exfoliated nanoparticles but also a bifurcation of the cracks as well as an induced plastic deformation of the matrix [144]. A similar improvement was obtained by Tangthana-umrung et al. [145] for the K_C_ of epoxy resin composites containing PES, graphene nanoplatelets (GNPs) and carbon nanotubes (CNTs). The addition of 5 wt% PES improved the mechanical properties and thermal stability including 35% and 40% increases in the tensile strength and fracture toughness, K_C_, respectively. Hybrid epoxy nanocomposites based on 5 wt% PES, GNPs and CNTs showed a synergistic toughening on the K_C_ toward the binary matrix systems, with a remarkably higher Young’s modulus in all epoxy/PES/CNT hybrid epoxy composites. The toughening of the GNP/PESU hybrid epoxy composites was attributed mainly to crack deflection by the PESU and GNP particles, as well as crack bifurcation by the GNP clusters.

If PSU and PES with similar molecular structures were used for their relatively high rigidity and glass transition temperatures, polyetherimide, poly(ether ether ketone) and acrylonitrile–butadiene–styrene terpolymer have been widely used due to the excellent combination of physical and mechanical properties including toughness, stiffness, chemical and solvent resistance as well as high glass transition temperature.

Studies on epoxy resin toughening using polyetherimide (PEI) combined with amino-grafted multiwalled carbon nanotubes (NH_2_–MWCNTs) or carboxyl-functionalised (COOH–MWCNTs) have been carried out by Chen al. [146] and Ma et al. [147], respectively. The results indicated a significant improvement in the fracture toughness, impact strength and flexural strength of the epoxy composites by the simultaneous incorporation of a low content of PEI and NH_2_–MWCNTs. Ternary composites with 5 phr PEI and 0.4 phr NH_2_–MWCNTs showed the maximum improvement with a synergistic effect in the fracture toughness, K*_C_* [146]. However, the critical strain energy release rate, G_C_, referred to as the fracture toughness of composites based on (COOH–CNTs), significantly increased, while the glass transition temperature and the storage modulus remained unaffected [147]. The interactions between the nano- and microscale toughening under appropriate mixing conditions led to a synergistic improvement in the G_C_ of the matrix. The debonding of the nanotubes and the resulting increase in the crack path can explain the toughening of the epoxy matrix. Hydroxyl-terminated poly (ether ether ketone) oligomer with pendant methyl groups (PEEKMOH) was used by Asif et al. [148] to toughen DGEBA epoxy resin modified with organically modified montmorillonite (oMMT). The fracture toughness, K_C_, and strain at break of epoxy composites based on 1 phr oMMT and 5 phr PEEKMOH increased by 65% and 85%, respectively, compared to the neat matrix, without synergistic reinforcement. This could be related to gel formation during the curing of the matrix.

Acrylonitrile–butadiene–styrene terpolymer (ABS) is a high-performance commercial thermoplastic with a specific structure, a rubbery polybutadiene dispersed in a rigid styrene–acrylonitrile copolymer system. Due to its excellent mechanical and thermal properties, ABS was used with nanoparticles to improve the properties of epoxy resin by several authors [149,150]. Mirmohseni and Zavareh [149] confirmed the increase in the tensile strength, strain at break and impact strength of a DGEBA resin with the addition of ABS and organically modified clay (Cloisite 30B). Adding 2.5 wt% clay and 4 phr ABS into the epoxy matrix resulted in a 133% increase in the impact strength. The impact and tensile strengths of the ternary composite were higher than the values of the binary systems, due to a good dispersion of the exfoliated clay platelets in the epoxy/ABS blend and crack blocking by ABS nanoparticles. A similar improvement in the same mechanical properties and associated mechanisms was observed in another study with hybrid epoxy composites containing ABS combined with multiwalled carbon nanotubes (MWCNTs) and diaminodiphenylsulfone as the matrix curing agent [150]. The impact strength (IS) and tensile strength (TS) increased significantly by 110% compared to the binary matrix system and 400% compared to the virgin matrix, while the TS was 165% higher than the binary system. The detachment of microparticles and nanoparticles can be used with the resulting energy to explain the improvement in the properties of the hybrid composites, as shown in Figure 8.

To further improve the properties, the epoxy resin was modified with methyl phenyl silicone (MPS), before the incorporation of ABS and graphene oxide (GO) [151]. The tensile strength of the grafted epoxy resin containing 5 wt% MPS, 2 wt% ABS and 0.1wt% GO increased by approximately 35% compared to the sample containing 5 wt% MPS and 2 wt% ABS, and up to 50% compared to the pure matrix. Morphology analysis of the samples revealed the presence of micro-sized domains which were obtained by phase separation, thus explaining the improvement in the properties.

Polycarbonate (PC) is an amorphous engineering thermoplastic which has good thermal stability, outstanding impact resistance and good transparency. The maximum improvement in the impact strength (IS), fracture toughness (K_C_) and flexural strength was obtained with an epoxy composite based on 1 wt% montmorillonite (Cloisite 30B) and 5 wt% by weight of PC [152]. An increase of 100% was obtained for the IS and K_C_, while the flexural strength was slightly improved compared to the unmodified epoxy resin. Hybrid composites containing nanofillers in combination with another thermoplastic not only showed a higher impact strength and resistance to crack propagation, but also showed a synergistic effect with respect to the fracture energy [24]. From analysis of the morphology and structure, it appeared that the toughening resulted in part from the extensive yielding of the matrix and the chemical reactions that took place between the epoxy resin and the polymeric modifier, which was partially solubilized in the resin (Figure 9).

Although polyamide (PA) is of great interest as a compatible thermoplastic reinforcing agent for brittle epoxy resin and because of the possible formation of chemical bonds, there have not been enough research studies of its use in hybrid epoxy nanocomposites. Gul et al. [153] conducted an extensive review on epoxy/polyamide modified with nanofillers, including their application and future trends. It has been shown that the properties of epoxy/polyamide nanocomposites depend on the type of components as well as on the level of dispersion of the inorganic nanoparticles. White and Sue [154] evaluated the mechanical and electrical properties of a successfully prepared epoxy/MWCNT composite reinforced with preformed PA particles and using an ultrasonic and solvent evaporation method. The ternary composite showed significant improvements in the mechanical properties of the matrix, including a synergistic effect on the tensile strain at break, the fracture toughness parameters, K_C_ and G_C_. This was attributed mainly to crack pinning and crack deflection in the vicinity of the large aggregates. However, the synergistic effect was not obtained with the composite of epoxy resin modified with polyamide 6 (PA6) and graphene oxide (GO) nanosheets grafted with PA6 [155]. The fracture toughness of the EP/PA6/GO increased by 52.6% compared to the neat epoxy resin, due to a better particle/matrix interface and rougher surface. A similar increase was obtained for the tensile strength and stiffness of an epoxy composite containing 5 phr of poly(styrene-co-acrylonitrile) and 1 wt% of MMT (Cloisite 20A), due to a good dispersion and reinforcement of nanoparticles [156]. High impact polystyrene (HIPS), as a standard thermoplastic material with a high impact strength, was combined with Cloisite 30B to improve the properties of the epoxy resin [157]. A significant improvement was reached in the tensile, compression and impact resistance of 60%, 65% and 400%, respectively, at the optimized modifier content. In addition, the tensile, flexural and strain at break under compression were improved by up to 55%, 40%, and 25% compared to the neat matrix, respectively, while the flexural strength remained unchanged. Fereidoon et al. [158] obtained a similar improvement in the previously mentioned properties when using multiwalled carbon nanotubes (MWCNTs) simultaneously with HIPS. The tensile, compressive and impact strength of the epoxy resin increased by 52%, 43% and 334%, respectively, while the tensile, flexural and compressive strain at break increased by 223%, 36% and 26%, respectively. Concurrent use of HIPS and MWCNTs resulted in synergistic effects on the ultimate tensile strength and tensile and flexural strain at break, most likely due to the homogeneous dispersion of the modifiers which act as crack stoppers and reinforcements of the matrix. Due to its high mechanical and thermal resistance, polyethylene terephthalate (PET) was combined with nanoparticles to toughen the epoxy resins. Prepared epoxy/PET/Al_2_O_3_ nanocomposites showed an improved tensile and impact strength combined with an excellent dielectric and heat resistance [159]. The improved properties of the ternary composites were remarkably superior to those of both the binary systems and the unmodified epoxy matrix. Sánchez-Cabezudo et al. [160] used polyvinyl acetate (PVA) for its miscibility with diglycidyl ether of bisphenol A resin (DGEBA) and two organomodified montmorillonites (Cloisite 30B and Cloisite 93A) to strengthen the polymer matrix. The ternary epoxy nanocomposites developed different morphologies depending on the PVAc content. As expected, the inclusion of solid nanoparticles in the epoxy matrix improved the stiffness but decreased the ductility, while PVA improved the toughness and reduced the stiffness. The tensile strength in the epoxy composites with Cloisite 30B was higher than with Cloisite 93A due to the stronger adhesion between the components. Ternary composites containing 10 wt% Cloisite 30B and 10 wt% PVAc exhibited a good balance between the tensile modulus and toughness. The most dominant toughening mechanism in hybrid epoxy/thermoplastic nanocomposites remains particle debonding from the matrix as well as crack deflection and matrix plastic deformations. Table 3 shows epoxy nanocomposites modified with thermoplastics showing the synergistic effects on selected matrix properties.

### 2.4. Hybrid Epoxy Nanocomposites Based on Block Copolymers

Block copolymers offer several advantages over other copolymers due to their specific structures which consist of blocks of chemically dissimilar homopolymers linked together by covalent bonds. The effectiveness of a small amount of block copolymers as toughening agents for epoxy resins has been confirmed by various researchers [161,162,163]. They have also been considered as potential reinforcing agents in combination with solid nanoparticles for epoxy resins. Block copolymers such as polystyrene–polybutadiene–polymethylmethacrylate (SBM) and polymethylmethacrylate–polybutylmethacrylate– polymethylmethacrylate (MAM) have been used as effective soft toughening agents of bisphenol A (DGEBA) epoxy resin [162,163]. Tao et al. [162] demonstrated that the fracture toughness parameter, K_C_, and impact strength of an epoxy resin containing 10 phr poly(methyl methacrylate)-*b*-poly(butyl acrylate)-*b*-poly(methyl methacrylate) triblock copolymer increased by 91.5% and 83.5%, respectively. They explained the increase in the mentioned properties by the nanophase structure obtained by self-assembly during the curing process. It was further confirmed that the toughness was improved to a large extent without significantly decreasing the glass transition temperature of the blends. In another work, the addition of 5 wt% poly(ethylene-*alt*-propylene)-*b*-poly(ethylene oxide) (PEP-PEO) (BCP) block copolymer to a DGEBA epoxy resin led to a 100% improvement in the fracture toughness (K_C_), without affecting the modulus and Tg [163]. The self-assembly of BCP molecules into well-dispersed wormlike micelles, which are more efficient than spherical micelles, also contributes to the K_C_ increase. As in most of the works previously described [85,94,110,139,144,145], the improvement in the properties was explained by the crack tip blunting, crack bridging, particle debonding and shear yielding of the matrix. When this BCP was incorporated with silica nanoparticles in the epoxy formulation, the BCP micelles were fixed on the surfaces of the nanoparticles, thus limiting the agglomeration of the latter [164]. The addition of 10 wt% nanosilica resulted in an increase in the critical stress intensity factor, K_C_, by 20% and the critical strain energy release rate, G_C_, by 25%, while the use of only 5 wt% BCP resulted in a significant improvement in the K_C_ (115%) and G_C_ (∼400%), compared to the virgin epoxy matrix. Moreover, the K_C_ and G_C_ parameters were further improved when simultaneously incorporating 10 wt% nanosilica and 5 wt% BCP into the epoxy resin, beyond those obtained with the EP/nanosilica systems and EP/BCP, individually. However, various investigations have confirmed that a synergistic improvement is obtained in the properties of hybrid epoxy nanocomposites when specific conditions are met. Jojibabu et al. [165] investigated the effect of functionalized carbon nanotubes (fCNTs) with two different triblock copolymers designated as SBM and MAM on the lap shear strength (LSS) of an epoxy resin. The LSS of composites based on 1 wt% fCNTs and 10 wt% MAM or 1 wt% fCNTs with 10 wt% SBM was 81% and 137% higher than that of the pure matrix. However, the highest LSS with a positive contribution of both modifiers was obtained when 0.5 wt% fCNTs were used in the ternary composite. It has been established that the increase in the LSS was caused by the excessive plastic deformation of the matrix in addition to the formation of voids and nanoparticle pull-out.

The addition of 5 phr polystyrene–polybutadiene–poly(methyl methacrylate) triblock copolymers with unfunctionalized carbon nanotubes (CNTs) resulted in an increase in the fracture toughness, K_C_, of 115% and critical energy release rate, G_C_, of 410% compared to the unmodified matrix [166]. Interestingly, the incorporation of a small amount, 0.25 phr, of CNTs also increased the fracture toughness parameters but only by 14.5%. Nevertheless, the combination of copolymers and CNTs did not further increase the fracture toughness of the rubber-modified epoxy, due to the small contribution of crack deformation and matrix yielding, which are considered along with particle pull-out as processes leading to polymer toughening. Schuster and Coelho [167] were also unable to obtain a positive toughening effect from the addition of a block copolymer (BCP), carbon nanotubes (CNT) and graphene nanoplatelets (GNP) on the properties of the epoxy resin. The results showed that a hybrid composite based on 0.5 wt% BCP, 0.25 w% CNT and 0.25 wt% GNP showed an increase in the K_C_ of 34% compared to the neat epoxy. The fracture toughness of the hybrid composite was not significantly improved compared to the binary epoxy systems, due to the difference in the mechanisms for the improvement in the properties and the inadequate proportion of the hybrid composite constituents. It seemed that the proportion of nanoparticles in the composite was not optimal to achieve a synergistic reinforcement of the matrix.

A similar trend was observed when a functionalized PMMA-bloc-PbuA-bloc-PMMA copolymer was used in combination with MWCNT nanofillers to develop hybrid epoxy nanocomposites [168]. The fracture toughness parameters, *K_C_* and *G_C_*, of the epoxy composite containing 6 wt% copolymer and 0.075 wt% MWCNTs increased from 0.57 to 1.41 MPa.m^1/2^ and 0.096 to 0.59 kJ/m^2^, respectively, but with no synergistic toughening effect. On the other hand, the tensile strength, strain at yield and modulus decreased with the increasing amount of copolymer particles, due to the softness of the particles. On the other hand, Zhang et al. [169] have recently demonstrated that the simultaneous addition of block copolymers based on polyethylene glycol and polypropylene glycol and multiwalled carbon nanotubes (MWCNTs) to an epoxy matrix resulted in a synergistic effect on the critical stress intensity factor (K_C_). The fracture parameter maximally increased with the introduction of 5 wt% copolymer or 0.25 wt% MWCNTs, compared to the neat matrix. With the simultaneous incorporation of 5 wt% copolymer and 0.25 wt% MWCNTs, the K_C_ value was significantly increased and was much higher than the binary system EP/copolymer and EP/MWCNTs. This was generated by a very good dispersion of the MWCNTs in the matrix, favored by the presence of the block copolymer, but also by the pull-out of MWCNT particles, crack bridging and matrix shear yielding. This is consistent with the positive results of Li et al. [170], who prepared ternary epoxy composites by using amine-terminated poly(butadiene−acrylonitrile)-functionalized graphene oxide (fGO) and a poly(ethylene oxide)-b-poly(ethylene-alt-propylene) diblock copolymer (BCP). The toughening efficiency of fGO alone was found to be highly dependent on the modifier concentration and the crosslink density of the matrix. The addition of 0.04 wt% GO with a lightly crosslinked density resulted in a 1.7-fold increase in the fracture energy, G_C_, over the neat epoxy, while the incorporation of 5 wt% BCP enhanced the parameter, G_C_, of the epoxy resin by a factor of 12. The combination of the rigid fGO and soft BCP particles at 0.04 wt% and 5 wt%, respectively, led to an 18 times increase in the G_C_ over the unmodified matrix and a 31% improvement over the binary systems.

A synergistic effect was also confirmed in the study on epoxy toughening by the simultaneous inclusion of block copolymers (BCPs) and core–shell particles (CSPs). It was shown that the epoxy/BCP/CSP flexural strength and fracture toughness (*K_C_*) were significantly enhanced compared to the neat epoxy, BCP/epoxy and CSP/epoxy systems, while the thermal stability was maintained almost unchanged [171]. The morphological analysis of the epoxy composites indicated that the BCP micelles were deformed and pulled out from the matrix, thus creating voids in the composites. These successive processes were considered responsible for the drastic improvement in the fracture toughness. Epoxy nanocomposites modified with block copolymers showing synergistic effects are shown in Table 4.

### 2.5. Hybrid Epoxy Nanocomposites Based on Diluents

It was reported that the inclusion of soft modifiers such as plasticizers or reactive diluents to epoxy resin generally deteriorate the mechanical properties, including the yield strength, stiffness and glass transition temperature. Plasticization of the epoxy matrix has also been considered as a possible method to increase some properties of epoxy resin by the flexibilization resulting from the increased free volume. Recently, reactive diluents have been considered as a potential modifier to improve the elasticity and impact strength of brittle epoxy resins. Rahman et al. [172] showed that the flexural strength, flexural modulus and energy to break of DGEBA epoxy resin increased maximally with 0.3 wt% NH_2_-MWCNTs (amino-functionalized multiwalled carbon nanotubes) and 10 phr polyether polyol. Furthermore, the hybrid nanocomposite showed a synergistic effect on the K_C_ and the flexural fracture energy of the epoxy matrix. The crack deflection, void formation with particle cavitation, nanoparticle debonding and induced shear yielding result in a drastic increase in the properties of the matrix [173]. A study carried out by Yi et al. [174] confirmed a positive toughening on the properties of an epoxy resin modified with oxidized multiwalled carbon nanotubes (oCNTs) and NCO-terminated reactive oligomer. The highest impact strength was obtained for the epoxy composite containing 10 wt% of oligomer and 0.5 wt% of oCNTs at room temperature (RT) and cryogenic temperature (CT), compared to the unmodified matrix as well as binary epoxy systems. The epoxy was toughened at RT and CT by the introduction of the flexible oligomer and was further reinforced by the addition of the oCNTs, without altering its tensile strength, due to positive interactions between the nanoclay and epoxy matrix as facilitated by the oligomer. In the case of the use of oxidized and amine-functionalized multiwalled carbon nanotubes (MWCNTs) with 5.0 wt% of reactive diluent, the IS of the epoxy resin was improved by up to 260% compared to the pure matrix, but no synergism [175]. However, a different result was obtained with the addition of oMMT (Cloisite 30B) and 7 wt% of polyether polyol which caused a decrease in the tensile and impact strength of the epoxy matrix with an increasing amount of nanoparticles [176].

Due to the presence of flexible ether linkages, polyethylene glycol (PEG) is expected to increase the toughness of epoxy resin. Chozhan et al. [177] reported that epoxy nanocomposites modified with PEG and/or octanediol (OCT) exhibited improved mechanical properties. The impact and tensile strengths of epoxy composites containing 3 wt% organoclay and 20 wt% PEG or 3 wt% organoclay and 10 wt% OCT were increased by 95% and 30%, respectively, compared to the neat epoxy. However, a considerable improvement was noted in the fracture toughness parameter, K_C_, of 255% and 334%, respectively, compared to the neat matrix when the epoxy matrix was modified with 0.1 wt% graphene oxide (GO) or 0.1% wt% of (PEG-grafted GO (GO-g-PEG). Moreover, the thermal degradation temperatures of epoxy nanocomposites based on GO and PEG-grafted GO have been increased by 50 °C [178].

Furthermore, Chang et al. [179] studied the effects of different processing conditions, diluent and nanosilica content on the properties of epoxy nanocomposites. As expected, the addition of a diluent reduced the hardness of the composites, while the impact strength of composites containing 1 wt% of nanopowders without a diluent increased compared to the neat epoxy matrix. Although the addition of a diluent made mixing processes easier and more efficient, it reduced the impact strength value of the nanocomposites. Diluents are also used to provide a uniform dispersion of nanoparticles in the polymer matrix and therefore contribute to improved epoxy performance properties [180].

The effect of hexanediol diglycidylether as a reactive diluent for an epoxy resin cured with an anhydride hardener and modified with silica nanoparticles (SNs) and core–shell rubber (CSR) nanoparticles was studied by Carolan et al. [181]. The addition of 25 wt% of a reactive diluent to the epoxy matrix resulted in a 38% increase in the fracture energy compared to a six-fold increase with the incorporation of 20 wt% of CSR nanoparticles without a diluent. However, the fracture energy (G_C_) improvement was much more significant with 25 wt% of a reactive diluent and 16 wt% CSR nanoparticles with a synergistic effect on the G_C_. The hybrid epoxy nanocomposites with both SNs and CSR showed only a modest increase in the G_C_. The increase in the toughness of the hybrid materials was related to the good dispersion of the nanoparticles in the epoxy polymer matrix. Another study concerned the contribution of 1,6-hexanediol diglycidyl ether (reactive diluent) in the toughening of DGEBA epoxy resin modified with graphene nanoplatelets (GNPs) and carboxyl-terminated acrylonitrile butadiene rubber (CTBN) or core–shell rubber (CSR) [182]. The fracture toughness parameter, K_C_, of the matrix increased by approximately 430% with the incorporation of 9 wt% CTBN and 12.5% of the reactive diluent. In addition, the incorporation of a reactive diluent has been shown to improve the toughness of the epoxy polymer modified with CSR particles, due to the increased ductility of the epoxy matrix. The corresponding toughening mechanisms are those described previously [172,173]. In the work undertaken by Sarafrazi et al. [183], thermoplastic polyester urethane (PU) was used to toughen a DGEBA epoxy resin containing copper chromite CuCr2O4 nanoparticles, castor oil (ECO) and montmorillonite (Cloisite 30B). A diminution in the tensile strength and rigidity was reported for the DGEBA/ECO system as a result of the decrease in the crosslinking density. Moreover, optimization of the properties revealed a synergistic toughening with the fracture toughness parameter (K_C_) and fracture energy (G_C_) of the DGEBA/ECO/Cloisite (at a ratio of 8:2:0.1) system which were found to be higher than the other epoxy systems. The improved hybrid properties have been explained mainly by the good dispersion of the nanoparticles as well as their interactions with the matrix, with the respective reactive groups. However, it was found that the use of epoxidized vegetable oil from *Mesua ferrea* L. seeds in an epoxy resin containing nanoparticles decreased the viscosity as expected but improved the mechanical and adhesive properties of the cured resin [184]. A significant enhancement in the adhesive strength was reached with 2.5 wt% nanoclay.

Liquid modifiers and solid nanoparticles are known to affect the properties of brittle epoxy resins differently, by flexibilization and strengthening, respectively. However, their simultaneous incorporation in the epoxy matrix can lead to positive toughening [71,73,81,83]. Table 5 summarizes the effects of nanoparticles and diluents on the properties of epoxy resins with relevant references.

### 2.6. Hybrid Epoxy Nanocomposites Based on Thermosets

Unsaturated polyester (UP) resins, one of the most widely used thermosetting resins, have very good chemical and corrosion resistance, high thermal stability, good processability and low cost. Although cured UP resins are brittle with a low impact strength and poor resistance to crack propagation, they are expected to act as a toughening agent for epoxy resin. It has already been confirmed that oMMT improves the mechanical properties of the epoxy matrix through the exfoliation/intercalation processes [57,59]. Moreover, the introduction of unsaturated polyester into epoxy resin has been shown to improve its mechanical properties due to chain entanglement and network formation [185].

Various researchers used organomodified montmorillonite (oMMT) in combination with UP to toughen epoxy resins [186,187,188]. Chozhan et al. [186] modified epoxy resin (EP) with oMMT clays and unsaturated polyester (UP). The impact strength of an EP reinforced with 5 wt% oMMT and 10 wt% UP was maximally increased by more than 35% over that of the unmodified matrix. The tensile and flexural properties of the cited hybrid nanocomposites were also enhanced, due to the formation of intercalated nanocomposites and the formation of chain entanglements between UP and the epoxy matrix. The contribution of two types of oMMT nanoclays, namely Cloisite 30B and Nanomer, on the toughening of epoxy resin modified with UP was reported in another study [187]. The UP-toughened epoxy composite containing 1 wt% Cloisite 30B showed the optimum increase in the tensile strength (15%), tensile modulus (20%), flexural strength (10%) and flexural modulus (20%) compared to the epoxy/UP blend. In addition, the mechanical properties of Cloisite-based nanocomposites were superior to those of nanocomposites with Nanomer and the adhesion between the epoxy/UP and Cloisite was superior to that between the epoxy/UP and Nanomer. The very good dispersion and exfoliation of the nanoparticles within the epoxy/UP system, in addition to the creation of an IPN structure, lead to an increase in the mechanical properties. Similar property enhancements have been reported in another study [186]. Other researchers have reported similar results [188]. The use of mechanical mixing followed by an ultra-sonication process resulted in a significant improvement in the tensile, impact and shear strengths of an epoxy/UP blend modified with bentonite [188]. However, manual mixing alone led to a reduction in the tensile strength and tensile modulus of an epoxy/UP/oMMT hybrid composite [189].

Studies have demonstrated that the increase in properties can be achieved by an optimum mixing of composite constituents [190,191]. In the work on UP/EP systems containing amine-modified silica nanoparticles (SNs), the decrease in the tensile and flexural strengths of the epoxy matrix due to the addition of UP was overcome by adding nanoparticles [190]. The combination of high shear mechanical mixing followed by an ultra-sonication process led to improved properties through the formation of crosslinked networks. Furthermore, hybrid epoxy composites containing UP and amine-functionalized multiwalled carbon nanotubes (NH2-MWCNTs), which were prepared using ball milling and sonication processes for the efficient dispersion of the nanoparticles in the epoxy matrix, exhibited improved tensile and flexural properties [191]. Tg values were also increased to an appreciable level with the incorporation of nanoparticles into the UP/EP system. The obtained positive results were explained by the homogeneous morphologies of the EP//UP blend and the chemical reactions between the reactive groups of EP and nanoparticles. However, Le and Huang [192] compared the effects of multiwalled carbon nanotubes (MWCNTs) and graphene nanoplatelets (GNPs) on the mechanical properties of an epoxy/unsaturated polyester mixture at different ratios. As in previous investigations, it was confirmed that a small amount of added MWCNTs (1 wt%) or GNPs (0.2 wt%) resulted in a significant increase of about 85% in the tensile strength of the epoxy/polyester blend.

Vinyl ester and phenolic resins have also been tested as potential toughening agents for epoxy/nanoclay composites [193,194]. Vinyl ester resin was used by Chozhan et al. [193] for its improved strength and chemical resistance to prepare composites with an epoxy resin (EP) and montmorillonite (oMMT). The obtained results showed that the addition of 10 wt% vinyl ester oligomer (VEO) and 5 wt% oMMT to the EP improved the tensile, flexural and impact strengths by more than 30% compared to the unmodified EP. The tensile modulus and flexural modulus values followed the same trend. The improvement in the mechanical properties was attributed to the formation of chain entanglements between EP and VEO, by the exfoliation of nanoclays as well as the homogeneous dispersion of montmorillonite in the epoxy system. Due to the acceptable mechanical properties already reported on phenolic–epoxy networks [194], the incorporation of both phenolic resin and nanoclay should improve the properties of the epoxy matrix. The epoxy/phenolic blend modified with 2.5 wt% montmorillonite (Cloisite 30B) showed a maximum increase in the fracture toughness (70%) and Young’s modulus (30%) compared to the virgin matrix [195]. The improvement in the toughness was attributed to the crack pinning and deflection as well as the formation of the shear band in the matrix.

As in the case of unsaturated polyester, cyanate ester resin possesses attractive performance properties but exhibits poor flexibility and high brittleness, which limits its use in many areas. It is expected that a mixture of epoxy resin (EP) and cyanate ester (CE) reinforced with selected nanoparticles would result in improved mechanical properties. EP/CE blends reinforced with functionalized multiwalled carbon nanotubes (MWCNTs) have been prepared by Li et al. [196] and their properties evaluated at room temperature (RT) and liquid nitrogen temperature (77 K). The functionalization of the MWCNTs improved their dispersion in the matrix as well as the interfacial bonding with the blend. The mechanical properties were greatly improved compared to nanocomposites based on non-functionalized MWCNTs. The tensile strength of the nanocomposites at 0.5 wt% of MWCNT loading increased by 11.6% at RT and 18.3% at 77 K, relative to the neat matrix. This increase was attributed to the good dispersion and strong interfacial bonding between the functionalized MWCNTs and the cyanate ester/epoxy system. The tensile modulus of the nanocomposites followed the same trend as the tensile strength. All presented studies showed an improved but without the expected synergistic effect which is the main goal of each hybrid composite.

The studies indicated that the introduction of unsaturated polyester resin into the epoxy resin and further reinforcement with nanoparticles improved the mechanical and damping properties to an appreciable extent but with no reported positive toughening and synergistic effect.

## 3. Applications of Epoxy Hybrid Nanocomposites

Depending on the improved properties, the prepared hybrid epoxy nanocomposites can be used in various fields. Epoxy/rubber nanocomposites can used as adhesives, while those based on engineering thermoplastics can be intended for the fabrication of advanced composite materials in the building, automotive and aerospace industries. Epoxy/unsaturated polyester systems are employed as a matrix for high-performance composites for engineering applications. The mentioned hybrid epoxy nanocomposites can also be used in existing fields but under severe conditions.

## 4. Future Trends and Challenges

This review presents advances on the improvement in the properties of epoxy nanocomposites modified with different types of modifiers. It focused on the toughening of diglycidyl ether of bisphenol A (DGEBA) epoxy resin by using solid nanoparticles with other modifiers such as rubbers, thermoplastics, thermosetting resins and diluents. It has been demonstrated that the performance properties of hybrid nanocomposites depend not only on the type and amount of components, and their eventual interactions, but also on the processing conditions [197,198,199].

The major challenges to be met in the context of obtaining hybrid polymer composites with high-performance properties remain in the choice of compatible polymers and nanofillers, the appropriate processing method of the constituents as well as the appropriate functionalization and dispersion of the nanofillers. The search for eco-friendly or green polymer nanocomposites is of great importance for researchers because of the necessary protection of the environment. The use of biobased and/or biodegradable modifiers seems to be a logical path toward reducing the problems of recycling ever-increasing quantities of plastic products.

Nevertheless, due to the very convincing results recorded during the last decade with remarkable synergies on the mechanical properties of epoxy resin, it would be wise to explore the method of reinforcing epoxy resin with only nanofillers such as montmorillonite, graphene platelets and multiwalled carbon nanotubes [200,201,202,203].

## Data Availability

Not applicable.

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
