# Peer review of "Hybrid Epoxy Nanocomposites: Improvement in Mechanical Properties and Toughening Mechanisms—A Review"

_polymers, 2023, doi:10.3390/polym15061398_

Round 1

Reviewer 1 Report

The manuscript "Hybrid Epoxy Nanocomposites: Improvement of Mechanical Properties and Toughening Mechanisms - a Review" by Anita Białkowska et al., presents resent experimental results in improving the mechanical and thermal properties of Bisphenol A epoxy via the development of nanohybrids composed of different nanoadditives and other modifiers and/or their combination. The authors have done a lot of work in reviewing all literature and have written a very lengthy paper. However, I have two comments that I consider important in order the paper to be published.

1.     This manuscript has 205 references, a large number of which are Review papers, not to mention recent ones. The authors should justify the need for another review paper on the subject.

2.     Despite the huge effort of the authors, the way this manuscript is written makes it rather a collection of systems with their respective results and not a review paper where comparisons should be made and conclusions should be deduced so that a reader can learn certain things and understand what is the important parameters that lead in the change of the different properties. The manuscript, in its current form is full of percentages that no one can remember at the end. To my opinion this manuscript should be re-written and the authors should try to further summarize the effects of the various additives.      

Overall, I believe that this manuscript cannot be considered for publication in Polymers in its current state and it could be published only after major revision and only when the authors address the above-mentioned comments.

Author Response

Dear Reviewer

Thank You for your timeand your appropriate remarks.  We have responded point by point to your questions and recommendations.

We hope and wish that You will accept our changes.

Kind regards

prof. M. Bakar

Reviewer 2 Report

The authors presented a review of the literature on epoxy resin toughening. The review article is generally welcome, considering it reports an overview of the art state. Before publication some improvements must be made:

> Introduction. In the final part of the introduction, the authors must present the manuscript's contribution to the industrial and academic sectors. In addition, it present the differential of the manuscript compared to those in the literature. For example, need more articles on epoxy resin toughening mechanisms?

> Scheme 2. Please add transmission microscopy (TEM) images to show differences in morphology;

> Authors need to enrich the manuscript by adding graphs of literature results. Just request authorization for reproduction;

> Authors need to add illustrations showing epoxy resin toughening mechanisms;

> In future perspectives, please, the authors should comment on the difficulty of recycling and the advantage of using natural modifiers to toughen;

> I recommend adding a topic only with the application of the toughened epoxy resin. Also, add illustrations of the required properties for the reported application;

> A table should be added summarizing the mechanical properties of toughened epoxy systems;

Author Response

Dear Reviewer

Thank You for your time and your appropriate remarks.  We have responded point by point to your questions and recommendations. Unfortunately, se did not receive permission to use our published figuers,. However, we included figures of our unpublished results.

We hope  that You will accept our changes.

Kind regards

prof. M. Bakar

Round 2

Reviewer 1 Report

These are my final comments concerning the review of manuscript ID: polymers-2211423 entitled "Hybrid Epoxy Nanocomposites: Improvement of Mechanical Properties and Toughening Mechanisms - a Review" by Anita BiaÅ‚kowska et al., that was submitted for publication in Polymers and I had recently reviewed.

Although, I am afraid that the authors did not respond quite satisfactory to my comments, I have to acknowledge that they have tried to include some qualitative explanation of the described experiments and that overall, they have done a lot of work in reviewing all the existing literature. Therefore, I recommend the publication of this manuscript in Polymers.

Reviewer 2 Report

The authors accommodated the recommendations in the manuscript, improving the quality. The manuscript has merit for publication, being a good contribution to the literature.